# VivesDebate-Speech: A Corpus of Spoken Argumentation to Leverage Audio Features for Argument Mining

**Ramon Ruiz-Dolz**
Centre for Argument Technology
University of Dundee
Dundee DD1 4HN, United Kingdom
rruizdolz001@dundee.ac.uk

**Javier Iranzo-Sánchez**
VRAIN-MLLP
Universitat Politècnica de València
46022 València, Spain
jairsan@upv.es

## Abstract

In this paper, we describe *VivesDebate-Speech*, a corpus of spoken argumentation created to leverage audio features for argument mining tasks. The creation of this corpus represents an important contribution to the intersection of speech processing and argument mining communities, and one of the most complete publicly available resources in this topic. Moreover, we have performed a set of first-of-their-kind experiments which show an improvement when integrating audio features into the argument mining pipeline. The provided results can be used as a baseline for future research.

## 1 Introduction

The automatic analysis of argumentation in human debates is a complex problem that encompasses different challenges such as mining, computationally representing, or automatically assessing natural language arguments. Furthermore, human argumentation is present in different mediums and domains such as argumentative monologues (e.g., essays) and dialogues (e.g., debates), argumentation in text (e.g., opinion pieces or social network discussions) and speech (e.g., debate tournaments or political speeches), and domains such as the political (Haddadan et al., 2019; Ruiz-Dolz, 2022), legal (Poudyal et al., 2020), financial (Chen et al., 2022), or scientific (Al Khatib et al., 2021; Bao et al., 2022) among others. Thus, human argumentation presents a linguistically heterogeneous nature that requires us to carefully investigate and analyse all these variables in order to propose and develop argumentation systems which are robust to these variations in language. In addition to this heterogeneity, it is worth mentioning that a vast majority of the publicly available resources for argumentation-based Natural Language Processing (NLP) have been created considering text features only, even if their original source comes from speech (Visser et al., 2020; Goffredo et al., 2022).

This is a substantial limitation, not only for our knowledge on the impact that speech may directly have when approaching argument-based NLP tasks, but because of the significant loss of information that happens when we only take into account the text transcript of spoken argumentation.

In this work, we will focus on the initial steps of argument analysis considering acoustic features, namely, the automatic identification of natural language arguments. Argument mining is the area of research that studies this first step in the analysis of natural language argumentative discourses, and it is defined as the task of automatically identifying arguments and their structures from natural language inputs. As surveyed in (Lawrence and Reed, 2020), argument mining can be divided into three main sub-tasks: first, the segmentation of natural language spans relevant for argumentative reasoning (typically defined as Argumentative Discourse Units ADUs (Peldszus and Stede, 2013)); second, the classification of these units into finer-grained argumentative classes (e.g., major claims, claims, or premises (Stab and Gurevych, 2014)); and third, the identification of argumentative structures and relations existing between these units (e.g., inference, conflict, or rephrase (Ruiz-Dolz et al., 2021a)). Therefore, our contribution is twofold. First, we create a new publicly available resource for argument mining research that enables the use of audio features for argumentative purposes. Second, we present first-of-their-kind experiments showing that the use of acoustic information improves the performance of segmenting ADUs from natural language inputs (both audio and text).

## 2 The VivesDebate-Speech Corpus

The first step in our research was the creation of a new natural language argumentative corpus. In this work, we present *VivesDebate-Speech*, an argumentative corpus created to leverage audio features for

argument mining tasks. The *VivesDebate-Speech* has been created taking the previously annotated *VivesDebate* corpus (Ruiz-Dolz et al., 2021b) as a starting point.

The *VivesDebate* corpus contains 29 professional debates in Catalan, where each debate has been comprehensively annotated. This way, it is possible to capture longer-range dependencies between natural language ADUs, and to keep the chronological order of the complete debate. Although the nature of the debates was speech-based argumentation, the *VivesDebate* corpus was published considering only the textual features included in the transcriptions of the debates that were used during the annotation process. In this paper, we have extended the *VivesDebate* corpus with its corresponding argumentative speeches in audio format. In addition to the speech features, we also created and released the BIO (i.e., *Beginning*, *Inside*, *Outside*) files for approaching the task of automatically identifying ADUs from natural language inputs (i.e., both textual and speech). The BIO files allow us to determine whether a word is the ***Beginning***, it belongs ***Inside***, or it is ***Outside*** an ADU.

The *VivesDebate-Speech* corpus is, to the best of our knowledge, the largest publicly available resource for audio-based argument mining. Furthermore, combined with the original *VivesDebate* corpus, a wider range of NLP tasks can be approached taking the new audio features into consideration (e.g., argument evaluation or argument summarisation) in addition to the standard argument mining tasks such as ADU segmentation, argument classification, and the identification of argumentative relations. Compared to the size of the few previously available audio-based argumentative corpora (Lippi and Torroni, 2016; Mestre et al., 2021) (i.e., 2 and 7 hours respectively), the *VivesDebate-Speech* represents a significant leap forward (i.e., more than 12 hours) for the research community not just in size (see Table 1) but also in versatility. A recent study (Mancini et al., 2022) explored also multimodal argument mining, but focusing exclusively on the classification of arguments and the detection of relations. A fine-grained approach to the segmentation of ADUs is not considered mostly because of the limitations of the existing resources for audio-based argument mining. This is an important limitation, given that audio features have an explicit role in spoken argumentation: argument delimitation. This is achieved, for example,

Table 1: Set-level statistics of the VivesDebate-Speech corpus. Each debate is carried out between two teams, and two to four members of each team participate as speakers in the debate.

| Set | # debates | Duration | # tags | | |
| --- | --- | --- | --- | --- | --- |
| | | | **B** | **I** | **O** |
| Train | 23 | 9.8h | 4605 | 63305 | 21432 |
| Dev | 3 | 1.3h | 692 | 8058 | 3752 |
| Test | 3 | 1.3h | 640 | 8413 | 3102 |

by changing the intonation and with the use of pauses. The *VivesDebate-Speech* is released under a Creative Commons Attribution-NonCommercial-ShareAlike 4.0 International license (CC BY-NC-SA 4.0) and can be publicly accessed from Zenodo[1].

## 2.1 Text

The text-based part of the *VivesDebate-Speech* corpus consists of 29 BIO files where each word in the debate is labelled with a BIO tag. This way, the BIO files created in this work enable the task of automatically identifying argumentative natural language sequences existing in the complete debates annotated in the *VivesDebate* corpus. Furthermore, these files represent the basis on which it has been possible to achieve the main purpose of the *VivesDebate-Speech* corpus, i.e., to extend the textual features of the debates with their corresponding audio features.

We created the BIO files combining the transcriptions and the ADU annotation files of the *VivesDebate*[2] corpus. For that purpose, we performed a sequential search of each annotated ADU in the transcript file of each corresponding debate, bringing into consideration the chronological order of the annotated ADUs.

## 2.2 Speech

Once the revised transcription has been augmented with the ADU information, the transcription was force-aligned with the audio in order to obtain word level timestamps. This process was carried out using the hybrid DNN-HMM system that was previously used to obtain the VivesDebate transcription, implemented using TLK (del Agua et al., 2014). As a result of this process, we have obtained *(start,end)* timestamps for every *(word,label)* pair. We split the

---

[1] https://doi.org/10.5281/zenodo.7102601
[2] https://doi.org/10.5281/zenodo.5145655

corpus into train, dev, and test considering the numerical order of the files (i.e., Debate1-23 for train, Debate24-26 for dev, and Debate27-29 for test). The statistics for the final *VivesDebate-Speech* corpus are shown in Table 1.

# 3 Problem Description

We approached the identification of natural language ADUs in two different ways: (i) as a token classification problem, and (ii) as a sequence classification problem. For the first approach, we analyse our information at the token level. Each token is assigned a BIO label and we model the probabilities of a token belonging to each of these specific label considering the $n$-length closest natural language contextual tokens. For the second approach, the information is analysed at the sentence level. In order to address the ADU identification task as a sequence classification problem we need to have a set of previously segmented natural language sequences. Then, the problem is approached as a 2-class classification task, discriminating argumentative relevant from non-argumentative natural language sequences.

# 4 Proposed Method

The use of audio information for argument mining presents significant advantages across 3 axes: efficiency, information and error propagation. Firstly, the segmentation of the raw audio into independent units is a pre-requisite for most Automatic Speech Recognition (ASR) system. If the segmentation produced in the ASR step is incorporated into the argument mining pipeline, we remove the need for a specific text-segmentation step, which brings significant computational and complexity savings. Secondly, the use of audio features allows us to take into account relevant prosodic features such as intonation and pauses which are critical for discourse segmentation, but that are missing from a text-only representation. Lastly, the use of ASR transcriptions introduces noise into the pipeline as a result of recognition errors, which can hamper downstream performance. Working directly with the speech signal allows us to avoid this source of error propagation.

Two different methods to leverage audio features for argument mining are explored in this paper. First, a standard end-to-end (E2E) approach that takes the text-based transcription of the spoken debate produced by the ASR as an input, and

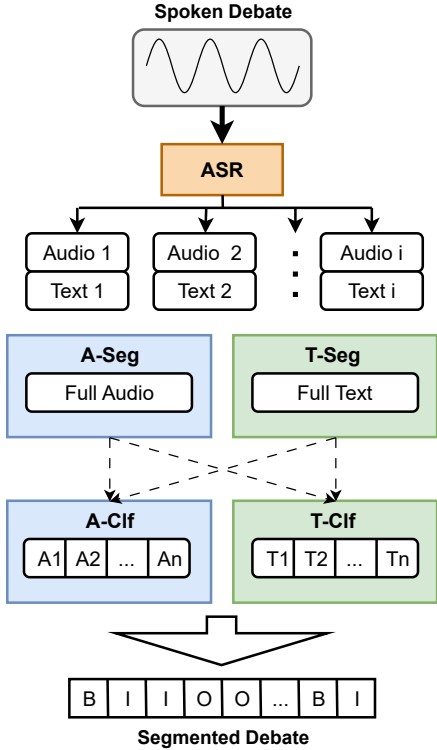

Figure 1: Overview of the proposed cascaded approach.

directly outputs the segmentation of this text into argumentative units. Second, we propose a cascaded model composed of two sub-tasks: argument segmentation and argument classification. In the first sub-task, the discourse is segmented into independent units, and then for each unit it is determined if it contains argumentative information or not. Both approaches produce an equivalent output, a sequence of BIO tags which is then compared against the reference. This work investigates how audio features can be best incorporated into the previously described process. An overview of the proposed cascaded method is shown in Figure 1. As we can observe, a segmentation step follows the ASR step, which segments either the whole audio (A-Seg) or the whole text (T-Seg) into (potential) argumentative segments. A classification step then detects if the segment contains an argumentative unit, by using either the audio (A-Clf) or the text (T-Clf) contained in each segment. If efficiency is a major concern, the segmentation produced by the ASR step can be re-used instead of an specific segmentation step tailored for argumentation, but this could decrease the quality of the results. This process significantly differs from the E2E approach where the BIO tags are directly generated from the output of the ASR step. This way, our cascaded

model is interesting because it makes possible to analyse different combinations of audio and text features.

The cascaded method has one significant advantage, which is that audio segmentation is a widely studied problem in ASR and Speech Translation (ST) for which significant breakthroughs have been achieved in the last few years. Currently, one of the best performing audio segmentation methods is SHAS (Tsiamas et al., 2022), which uses a probabilistic Divide and Conquer (DAC) algorithm to obtain optimal segments. Furthermore, we have compared SHAS with a Voice Activity Detection (VAD) baseline, as well as with the non-probabilistic VAD method (Potapczyk et al., 2019) using a Wav2Vec2 pause predictor (Baevski et al., 2020), which performs ASR inference and then splits based on detected word boundaries (see Appendix A.1). To complete our proposal, we have also explored text-only segmentation methods in which a Transformer-based model is trained to detect boundaries between natural language segments. This way, each word can belong to two different classes, boundary or not boundary.

The second stage of our cascaded method is an argument detection classifier, that decides, for each segment, if it includes argumentative content and should be kept, or be discarded otherwise. In the case that our classifier detects argumentative content within a segment, its first word is assigned the *B* label (i.e., *Begin*) and the rest of its words are assigned the *I* label (i.e., *Inside*). Differently, if the classifier does not detect argumentative content within a segment, all the words belonging to this segment are assigned the *O* label (i.e., *Outside*).

The code used to implement all the experiments reported in this paper can be publicly accessed from GitHub[3]. Furthermore, the weights of all the text-based models resulting from our experiments can be publicly downloaded from the Huggingface[4] repository.

## 5   Results

Once the hyperparameters of each individual model have been optimised considering the experimental setup defined in the Appendix A.2, the best results for each system combination are reported on Table 2. The results are consistent across the dev

[3]https://github.com/jairsan/VivesDebate-Speech

[4]E2E Model, Text Segmenter, and Text Classifier.

Table 2: Accuracy and Macro-F1 results of the argumentative discourse segmentation task on both *dev* and *test* sets.

| Model | Dev | | Test | |
|---|---|---|---|---|
| | Acc. | Macro-F1 | Acc. | Macro-F1 |
| *E2E BIO-5* | 0.71 | 0.45 | 0.72 | 0.47 |
| *E2E BIO-A* | 0.72 | 0.48 | 0.75 | 0.49 |
| *T-Seg+A-Clf* | 0.59 | 0.41 | 0.58 | 0.41 |
| *T-Seg+T-Clf* | 0.64 | 0.49 | 0.69 | 0.49 |
| *A-Seg+A-Clf* | 0.60 | 0.42 | 0.58 | 0.43 |
| *A-Seg+T-Clf* | 0.67 | **0.51** | 0.70 | **0.51** |

and test sets. The end-to-end model outputs the BIO tags directly, either from fixed length input (of which 5 was also the best performing value), denoted as *E2E BIO-5*. Alternatively, the *E2E BIO-A* was trained considering the natural language segments produced by the SHAS-multi model instead of relying on a specific maximum length defined without any linguistic criteria. This way, it was our objective to improve the training of our end-to-end model through the use of linguistically informed audio-based natural language segments. It can be observed how this second approach leverages the audio information to improve test macro-F1 from 0.47 to 0.49.

For the cascade model, we test both audio and text segmenters and classifiers. Similarly to the E2E case, the use of audio segmentation consistently improves the results. For the text classifier, moving from text segmentation (*T-Seg + T-Clf*) to audio segmentation (*A-Seg + T-Clf*) increases test macro-F1 from 0.49 to 0.51. Likewise, when using an audio classifier *(A-Clf)*, audio segmentation improves the test results from 0.41 to 0.43 macro-F1. However, the relatively mediocre performance of the audio classification models with respect to its text counterparts stands out. We believe this could be caused due to the fact that speech classification is a harder task than text classification in our setup, because the audio classifier deals with the raw audio, whereas the text classifier uses the reference transcriptions as input. Additionally, the pretrained audio models might not be capable enough for some transfer learning tasks, as they have been trained with a significantly lower number of tokens, which surely hampers language modelling capabilities.

## 6 Conclusions

We have presented the *VivesDebate-Speech* corpus, which is currently the largest speech-based argument corpus. Furthermore, the experiments have shown how having access to audio information can be a source of significant improvement. Specifically, using audio segmentation instead of text-based segmentation consistently improves performance, both for the text and audio classifiers used in the cascade approach, as well as in the end-to-end scenario, where audio segmentation is used as a decoding constraint for the model.

## Limitations

The goal of the experiments presented in this work is simply to highlight some of the advantages that can be gained by integrating audio information into the argumentation pipeline, but this raises questions about how should evaluation be carried out. In this paper we have simulated an ASR oracle system in order to be able to compare the extracted arguments against the reference. We have reported accuracy and F1 results by assuming that this is a standard classification task, and therefore incorrect labels all incur the same cost. However, there is no definitive proof that this assumption is correct. Given that argumentative discourse is a highly complex topic that relies on complex interactions between claims, a more realistic evaluation framework might be needed. Something that warrants further study, for example, is whether precision and recall are equally important, or if it is better for example to have higher recall at the expense of precision. In particular, the comprehension of an argument might not be hampered by additional or redundant ADUs, but might be significantly harder if a critical piece of information is missing.

Likewise, in order to move towards a more realistic setting, using a real ASR system instead of an oracle ASR system should be preferred. This could only be achieved by devising a new evaluation framework that can work even if the ASR system introduces errors, but it would provide a more accurate comparison between systems using audio or text features. For the comparison show in this work, the audio systems work with the raw audio, whereas the text systems use the oracle transcription. In a realistic setting, the errors introduced by the ASR system would make it harder for the text-based models, and therefore the gap between the two could be significantly bigger that what has been shown here.

## Acknowledgements

This work is partially supported by the Spanish Government project PID2020-113416RB-I00, by the Spanish Ministry of Economy and Competitiveness (MINECO) under reference no. TIN2015-68326-R (MORE), the FPU scholarship FPU18/04135, and by the 'AI for Citizen Intelligence Coaching against Disinformation (TITAN)' project, funded by the EU Horizon 2020 research and innovation programme under grant agreement 101070658, and by UK Research and innovation under the UK governments Horizon funding guarantee grant numbers 10040483 and 10055990.

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

# A Appendix

## A.1 Audio Segmentation Algorithms

Table 3 shows the best performance on the dev set of the different audio segmentation methods tested. Results are reported without using an argument classifier, which is equivalent to a majority class classifier baseline, as well as an oracle classifier which assigns the most frequent class (based on the reference labels) to a segment. This allows us to analyse the upper-bound of performance that could be achieve with a perfect classifier.

Table 3: Audio segmentation methods performance on the dev set, as measured by accuracy (Acc.) and Macro-F1.

| Method | Classifier | | | |
| --- | --- | --- | --- | --- |
| | Majority | | Oracle | |
| | Acc. | Macro-F1 | Acc. | Macro-F1 |
| Baseline (VAD) | 0.53 | 0.35 | 0.70 | 0.53 |
| DAC xlsr-53 | 0.61 | 0.34 | 0.81 | 0.65 |
| DAC xls_r-1b | 0.61 | 0.35 | 0.81 | 0.64 |
| SHAS-es | 0.64 | **0.37** | 0.83 | **0.66** |
| SHAS-ca | 0.64 | 0.36 | 0.82 | 0.64 |
| **SHAS-multi** | 0.65 | **0.37** | 0.83 | **0.66** |

The results highlight the strength of the SHAS method, with the SHAS-es and SHAS-multi models which are working on a zero-shot scenario, outperforms the Catalan W2V models. The SHAS-ca model had insufficient training data to achieve parity with the zero-shot models trained on larger audio collections. As a result of this, the SHAS-multi model was selected for the rest of the experiments.

One key factor to study is the relationship between the maximum segment length (in seconds) produced by the SHAS segmenter, and the performance of the downstream classifier. Longer segments provide more context that can be helpful to

the classification task, but a longer segment might contain a significant portion of both argumentative and non-argumentative content. Figure 2 shows the performance of the text classifier as a function of segment size, measured on the dev set. 5 seconds was selected as the maximum sentence length, as shorter segments did not improve results.

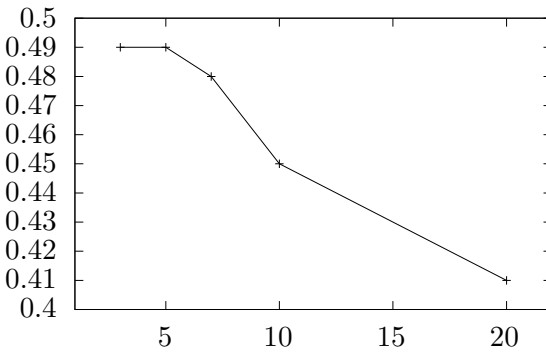

Figure 2: Dev set F1 score as a function of maximum segment length (s), SHAS-multi segmenter followed by text classifier.

## A.2 Experimental Setup

Regarding the implementation of the text-based sub-tasks (see Figure 1, green modules) of our cascaded method for argument mining, we have used a RoBERTa (Liu et al., 2019) architecture pre-trained in Catalan language data[5]. For the segmentation (T-Seg), we experimented with a RoBERTa model for token classification, and we used the segments produced by this model to measure the impact of the audio features compared to text in the segmentation part of our cascaded method. For the classification (T-Clf), we finetuned a RoBERTa-base model for sequence classification with a two-class classification *softmax* function able to discriminate between the argument and the non-argument classes. As for the training parameters used in our experiments with the RoBERTa models, we ran 50 epochs, considering a learning rate of 1e-5, and a batch size of 128 samples. The best model among the 50 epochs was selected based on the performance in the dev set. All the experiments have been carried out using an Intel Core i7-9700k CPU with an NVIDIA RTX 3090 GPU and 32GB of RAM.

The implementation of the audio-based sub-tasks (see Figure 1, blue modules) is quite differ-

ent between segmentation to classification. For audio-only segmentation (A-Seg), we performed a comparison between VAD, DAC, and SHAS algorithms. We evaluated the performance of these algorithms on the dev partition, and we selected the SHAS algorithm to be integrated into the cascaded architecture for the final evaluation of our proposal. For the hybrid DAC segmentation, two Catalan W2V ASR models are tested, xlsr-53[6] and xls-r-1b[7]. In the experiments, we used the original SHAS Spanish and multilingual checkpoints. Additionally, we trained a Catalan SHAS model with the *VivesDebate-Speech* train audios, as there exists no other public dataset that contains the unsegmented audios needed for training a SHAS model. Regarding our audio-only classifier (A-Clf), Wav2Vec2[8] models have been finetuned on the sequence classification task (i.e., argument/non-argument).

---

[5]projecte-aina/roberta-base-ca-v2 was used for all RoBERTa-based models reported in this work

[6]softcatala/wav2vec2-large-xlsr-catala
[7]PereLluis13/wav2vec2-xls-r-1b-ca
[8]facebook/wav2vec2-xls-r-300m and facebook/wav2vec2-xls-r-1b