# OpenReview forum: "VivesDebate-Speech: A Corpus of Spoken Argumentation to Leverage Audio Features for Argument Mining"
_EMNLP/2023/Conference — EMNLP 2023 Main_

### Official Review · Reviewer_s6XU · 2023-08-04

**Soundness:** 4

**Excitement:**

4: Strong: This paper deepens the understanding of some phenomenon or lowers the barriers to an existing research direction.

**Paper Topic And Main Contributions:**

This submission introduces an audio alignment for an argumentation corpus. The authors use the aligned audio for experimenting with non-textual features for argument labeling.  They find that using audio features improves segmentation for text-based classification, yielding (slightly) better annotations than when using text-based segmentation.

The two contributions (resources & experiments) are both interesting, but each of them lacking certain aspects.

For the resources, I would have liked to see a more detailed description, for example:
 - how many speakers are there? how long are the continuous segments per speaker (if you have speaker change)?
 - is the speaker identity annotated in the data?
 - how did you deal with incorrect alignments from the automatic alignment process? For resource papers, it is always important to me to give information about the data gathering process, because this information cannot be recovered later on by other researchers (in contrast to e.g. new experiments).
Some of this information can be found in the actual data, but it is in my opinion still good to directly report it.

For the experiments, there are two aspects:
 - the authors report accuracy and F1 score.  Their baseline model has the highest accuracy, but only the F1 score is discussed in the paper (where their own model has higher scores). It is not clear to me why a metric is reported but then completely ignored. As the accuracy paints a different picture than F1, it would be good to at least have a succinct explanation for this decision
 - it is not clear to me why the authors chose to report the accuracy of a pure text-based classifier and a pure audio-based classifier but not a classifier using both as input. I am not claiming that it needs to be in the paper but using all available features is probably what most people think about first and I would add a short explanation (could very well be "we are not interested in it", but better formulated)

I wrote these topics here instead of in "reasons to reject" because these are aspects where I would like to see more but at the same recognize that this is a focused short paper. If anything, it seems to me that the page restrictions are a bit too small for this content; I am e.g. also missing a more in-depth description of the classifiers you used in your experiments. Other than that, I enjoyed reading this submission.

**Questions For The Authors:**

A) Why did you not discuss the accuracy / why do think that F1 is the relevant metric?

**Reasons To Accept:**

- Provides a valuable resource
 - has an interesting experiment set, given the space restrictions

**Reasons To Reject:**

- the classification systems used in the experiments are not described, as far as I can see. There is only Fig 1 showing the data flow.

**Reproducibility:**

3: Could reproduce the results with some difficulty. The settings of parameters are underspecified or subjectively determined; the training/evaluation data are not widely available.

**Reviewer Confidence:**

4: Quite sure. I tried to check the important points carefully. It's unlikely, though conceivable, that I missed something that should affect my ratings.

---

> ### Author Rebuttal · Authors · 2023-08-27
>
> Dear reviewer s6XU,
>
> First of all we would like to thank you for your insightful feedback and constructive comments. In our response, we would like to address the mentioned reason to reject and the question with hopes of improving your impression in our work.
>
> Regarding the main concern and reason to reject about the lack of description of the classification models used in our paper, all the details are provided in the appendix. Appendix A.1 Audio Segmentation Algorithms and Appendix A.2 Experimental Setup contain all the details of the different algorithms and models that we used to implement the system depicted in Figure 1.
>
> Second, with respect to your question A), we focused our discussion on the macro F1 instead of accuracy because it is a less optimistic metric. In highly unbalanced datasets such as ours, macro F1 is more informative since a majority class classifier would be severally punished while with accuracy not. However, with the camera ready version of our paper we can extend the analysis of both metrics in the additional page.
>
> Finally, regarding your concerns about the resource description, we did not include the information of the speakers since it can be extended from the source VivesDebate corpus. Each debate has an average of 6 speakers (3 speakers per team) and each speaker has a complete continuous speech of 10-15 minutes. In our corpus there are a total of 16 teams, therefore we can estimate that 45-55 speakers were involved in these debates. However, the speaker identity was not included in any of the two corpora.
>
> Regarding your concern about the incorrect alignment, we would like to mention that the starting point for this extension are the human transcriptions of the VivesDebate corpus, so there is a 100% match between the text to be aligned and the words being said. The automatic alignment is just a matter of finding the correct timestamp for each word. This specific alignment task is sufficiently mature and we have been doing it for many years, so our models can perfectly align the audio with the text. No manual post-editing is applied after the alignment as we estimate that alignment accurary is close to 100% in this case (A manual revision of Debate 1 did not show any alignment mistakes, and we are confident that this also applies to the rest of the debates).
>
> We hope that with this response we will be able to improve your perception on the Soundness and Excitement of our work.
>
> Thank you very much again for your time and dedication,
>
> The authors.

---

### Official Review · Reviewer_hNqj · 2023-08-04

**Typos Grammar Style And Presentation Improvements:** None
**Soundness:** 4

**Excitement:**

4: Strong: This paper deepens the understanding of some phenomenon or lowers the barriers to an existing research direction.

**Missing References:**

None

**Paper Topic And Main Contributions:**

The papers presents a novel corpus of *spoken* argumentation from argument mining. It also presents results from an experiment using this corpus and showing that including speech features improves the predictive performance.

**Questions For The Authors:**

I enjoyed reading the paper very much. I have no question for the authors. I apologize to the authors if this is not valuable feedback.

**Reasons To Accept:**

The paper presents an interesting and useful corpus. It is well-written and overall it looks sound in technical terms.

**Reasons To Reject:**

No reason to reject.

**Reproducibility:**

4: Could mostly reproduce the results, but there may be some variation because of sample variance or minor variations in their interpretation of the protocol or method.

**Reviewer Confidence:**

3: Pretty sure, but there's a chance I missed something. Although I have a good feel for this area in general, I did not carefully check the paper's details, e.g., the math, experimental design, or novelty.

---

> ### Author Rebuttal · Authors · 2023-08-27
>
> Dear reviewer hNqj,
>
> We would like to thank you very much for your time and effort reviewing our work. We are also very pleased to read your kind words about our work.
>
> Thank you very much,
>
> The authors.

---

### Official Review · Reviewer_N8m1 · 2023-08-12

**Soundness:** 3

**Excitement:**

3: Ambivalent: It has merits (e.g., it reports state-of-the-art results, the idea is nice), but there are key weaknesses (e.g., it describes incremental work), and it can significantly benefit from another round of revision. However, I won't object to accepting it if my co-reviewers champion it.

**Paper Topic And Main Contributions:**

This paper proposes VivesDebate-Speech, which augments the VivesDebate corpus with spoken argumentation in an attempt to leverage audio features for argument mining tasks. VivesDebate-Speech represents a significant leap forward for the research community in terms of size and versatility. Compared to the few previously available audio-based argumentative corpora, VivesDebate-Speech provides more than 12 hours of spoken argumentation, which doubles the size of the existing largest audio-based corpus. The paper also highlights the advantages of integrating audio information into the argumentation pipeline, and provides a baseline for future research in this area

The authors point out that human argumentation presents a linguistically heterogeneous nature that requires careful investigation, which makes it reasonable to incorporate audio features into argumentative corpora. The creation of this corpus represents an important contribution to the intersection of speech processing and argument mining communities.

**Questions For The Authors:**

1) How do you deal with possible noise in audio, for example, mispronunciation or repeated words?

2) How does VivesDebate-Speech compare to other publicly available resources for audio-based argument mining, and what are the unique contributions of this corpus besides its size?

3) Why does the incorporation of audio features hurt the accuracy compared with E2E BIO (Table 2)?

**Reasons To Accept:**

1) The creation of a comprehensive and versatile corpus of spoken argumentation, which provides a valuable resource for researchers in the field of audio-based argument mining, as well as the potential for new insights and advancements in this field.

2) This paper highlights the advantages of integrating audio information into the argumentation pipeline, and provides a baseline for future research in this area.

**Reasons To Reject:**

1) The lack of comprehensive results from experiments performed using VivesDebate-Speech, such as using models of different structures and of various sizes.

**Reproducibility:**

3: Could reproduce the results with some difficulty. The settings of parameters are underspecified or subjectively determined; the training/evaluation data are not widely available.

**Reviewer Confidence:**

3: Pretty sure, but there's a chance I missed something. Although I have a good feel for this area in general, I did not carefully check the paper's details, e.g., the math, experimental design, or novelty.

---

> ### Author Rebuttal · Authors · 2023-08-27
>
> Dear reviewer N8m1,
>
> First of all we would like to thank you for your insightful feedback and constructive comments. In our response, we would like to address the mentioned reason to reject and the three questions with hopes of improving your impression in our work.
>
> We would like to clarify your concern, which have led you to give a unique reason for rejecting our work. This short paper has two main contributions. The first one is the creation of VivesDebate-Speech, the largest available resource for argument mining including audio information. The second one is a set of first-of-its kind experiments in which we are able to observe a small, but consistent improvement in the performance of argument mining systems when considering acoustic features in addition to text. Therefore, it was not our objective to provide a deep analysis of different algorithms and configurations, but to generate a new corpus and provide a set of solid baselines that can be used as reference in future work done in the direction of analysing or proposing new algorithms for the task of argument mining from speech.
>
> We would also like to give an answer to the three questions included in the review.
>
> 1. The speakers know the topic of the debate far in advance, and have received training in public speaking, so most of the speeches
> are well prepared and without any "noise" (in the sense of hesitations, repetitions, etc). In the very few cases where there might be issues, the annotators were instructed to remove any noise from the argumentative discourse units.
>
> 2. In addition to its size, the VivesDebate-Speech corpus expands the previous available speech-based argument mining corpora in means of its argumentative richness. Given the nature of the original VivesDebate corpus containing argumentative annotations of the complete lines of reasoning present in debates of an average length of 40 minutes, this argumentative richness is also reflected in the VivesDebate-Speech corpus. Making it one of the most argumentatively dense corpora publicly available.
>
> 3. The incorporation of audio features does not hurt the accuracy at all, in fact what we can observe in Table 2 is that combining audio features with text feature we can consistently improve the performance of our systems in both, the E2E and the cascaded approaches. It would not be fair, however, to directly compare the E2E with the cascaded approaches since both tasks are significantly different in nature (i.e., token labelling vs. sequence labelling) and in complexity (i.e., number of classes). See Sections 3 and 4.
>
> We hope that with this response we will be able to improve your perception on the Soundness and Excitement of our work.
>
> Thank you very much again for your time and dedication,
>
> The authors.

---

### Meta-Review · Area_Chair_HXG6 · 2023-09-18

**Recommendation:** 4

**Metareview:**

This paper proposes VivesDebate-Speech, which augments the VivesDebate corpus with spoken argumentation in an attempt to leverage audio features for argument mining tasks. The size (about double the size of a comparable audio dataset) and versatility of the corpus have shown to be very interesting for the research community, it offers the chance for AM and speech processing communities to work together, and it highlights the benefit of integrating audio features in computation argumentation tasks.

The reviewers generally agreed that the paper is well written and clear. The most significant issue brought up in the reviews was the lack of detail in the description of both the resource and the experiments (which the reviewer also sees is an issue related to space restrictions). The concern was outlined in detail and the authors thoroughly addressed each point in the author response. I believe their response should be sufficient to dampen the concerns brought up.

---

### Decision · Program_Chairs · 2023-10-07

**Decision:**

Accept-Main

**Comment:**

This paper proposes VivesDebate-Speech, which augments the VivesDebate corpus with spoken argumentation in an attempt to leverage audio features for argument mining tasks. The size (about double the size of a comparable audio dataset) and versatility of the corpus have shown to be very interesting for the research community, it offers the chance for AM and speech processing communities to work together, and it highlights the benefit of integrating audio features in computation argumentation tasks.

The reviewers generally agreed that the paper is well written and clear. The most significant issue brought up in the reviews was the lack of detail in the description of both the resource and the experiments (which the reviewer also sees is an issue related to space restrictions). The concern was outlined in detail and the authors thoroughly addressed each point in the author response. I believe their response should be sufficient to dampen the concerns brought up.